# Accelerometer-Measured Physical Activity, Inactivity, and Related Factors in Family Caregivers of Patients with Terminal Cancer

**DOI:** 10.3390/ijerph20010179

**Published:** 2022-12-22

**Authors:** Inmaculada Valero-Cantero, Cristina Casals, Juan Corral-Pérez, Francisco Javier Barón-López, Julia Wärnberg, María Ángeles Vázquez-Sánchez

**Affiliations:** 1Puerta Blanca Clinical Management Unit, Malaga-Guadalhorce Health District, 29004 Malaga, Spain; 2ExPhy Research Group, Department of Physical Education, University of Cadiz, Puerto Real, 11519 Cadiz, Spain; 3Biomedical Research and Innovation Institute of Cádiz (INiBICA) Research Unit, Puerta del Mar University Hospital, 11009 Cadiz, Spain; 4Department of Preventive Medicine, Public Health and Science History, University of Malaga, 29071 Malaga, Spain; 5Malaga Biomedical Research Institute (IBIMA), 29590 Malaga, Spain; 6Department of Nursing, Faculty of Health Sciences, University of Malaga, 29071 Malaga, Spain; 7PASOS Research Group, UMA REDIAS Network of Law and Artificial Intelligence Applied to Health and Biotechnology, University of Malaga, 29071 Malaga, Spain

**Keywords:** informal caregivers, sedentary behaviour, burden, strain, palliative, quality of life, lifestyle

## Abstract

The physical activity (PA) and inactivity of family caregivers of cancer patients were investigated and related to burden and quality of life through a cross-sectional multicentre study. A total of 75 caregivers were recruited from June 2020 to March 2021. The levels of PA and inactivity were estimated with a wrist accelerometer, 24 h a day, for 7 consecutive days. The Quality of Life Family Version, the Caregiver Strain Index, the total duration of care, the average number of hours spent in care, and the assistance received were registered. Our results showed that moderate-to-vigorous PA was 96.40 ± 46.93 min/day, with 90.7% of participants performing more than 150 min/week of physical activity, and this was significantly associated with age (r = −0.237). Daily inactivity was 665.78 ± 94.92 min, and inactivity for 20–30 min was significantly associated with caregiver burden (r = 0.232) and quality of life (r = −0.322). Compliance with the World Health Organization recommendations was significantly associated with a lower quality of life (r = −0.269). The strength of these associations was limited (r ~0.2). In conclusion, the PA performed by most caregivers met the established recommendations, although older caregivers (>65 years old) performed lower moderate-to-vigorous PA than younger ones. In addition, the mean inactive time was high (11 h/day), showing slight relationships with the burden and quality of life of caregivers.

## 1. Introduction

Informal caregivers of patients in home palliative care are usually family members who spend a large part of their day providing accompaniment, care, and attention. Indeed, the time dedicated to this care may approach 24 h a day, especially for caregivers who do not have paid work outside the home [1]. Moreover, palliative care is often maintained over many months, and even before this (terminal) stage, many caregivers have spent long periods providing for the person with cancer [2]. This situation can lead to social isolation resulting in physical inactivity [3].

The World Health Organization (WHO) currently recommends that adults should engage in at least 150 min per week of physical activity of moderate-to-vigorous intensity (MVPA). Previous WHO recommendations stipulated that aerobic activity should be performed in sessions lasting at least ten minutes. However, new evidence, obtained via electronic monitoring devices, shows that physical activity of any duration (i.e., with no minimum threshold) is also associated with a better health outcome [4].

Caregivers may experience physical harm of varying degrees of severity, especially among older caregivers, such as musculoskeletal disorders [5,6], pain [7], and weight gain [8]. Furthermore, they are at an increased risk of mental disorders, such as depression and anxiety, of caregiver burden, and worsening of the overall quality of life [9]. This is relevant to highlight that these outcomes are often related to the presence and intensity of the patient’s symptoms [10,11,12].

Regular physical activity is beneficial in many ways, including improved mental health [13] and reduced depression [14,15] and anxiety [16]. It also has cardiorespiratory benefits, contributing to the prevention of chronic non-communicable diseases such as cardiovascular disease [17] and has a protective effect against Alzheimer’s disease [18]. In contrast, physical inactivity and a sedentary lifestyle provide no such advantages and may provoke tangible harm [19,20,21].

Little is known about the physical activity of family caregivers and associated factors, especially in caregivers of patients with advanced cancer, and the relationship between the levels of physical activity and inactivity with the caregiver’s situation. A recent systematic review considered the contribution of a programme of physical exercise for the family caregivers of patients with chronic diseases, concluding that it achieved minor-to-moderate improvements in the caregivers’ mental and physical health. However, after eliminating the studies probably subject to bias, the observed improvement was no longer significant [22].

Hence, the aim of the present study was to evaluate the accelerometer-measured physical activity and inactivity time of caregivers of patients with advanced cancer in home palliative care and also to examine the relationship between the caregiver burden and quality of life.

## 2. Materials and Methods

### 2.1. Design

Cross-sectional descriptive study, carried out in six clinical management units in the context of primary healthcare in Andalusia, Spain. A sample of 75 participants was estimated as necessary using the Epidat 4.2 program for an alpha error of 0.05 and an estimation error of 0.5, according to previous research [23]. According to our main associations, the relationship between quality of life and sustained inactivity bouts of 20 to 30 min and with total MVPA, we obtained a computing-achieved power of 0.82 and 0.86, respectively, using the G*Power software (v. 3.1.9.6, University of Kiel, Germany).

### 2.2. Participants

The following inclusion criteria were applied: (i) family caregiver of a cancer patient receiving home palliative care; (ii) aged at least 18 years. The exclusion criteria were: (i) allergy to plastic or metal; (iii) not acting as a caregiver during the entire seven days of the study period. Potential study participants were identified from the patient lists of the Palliative Care Assistance Process.

The 75 caregivers participating in the study were recruited between June 2020 and March 2021. At the initial contact with the caregiver, the researcher concerned checked that all inclusion criteria were met. If so, full information about this study was given, and the caregiver’s signed informed consent to participate was requested. Each family caregiver was interviewed by a nurse in the patient’s home and asked to complete the study questionnaire and to wear the wrist accelerometer 24 h a day for seven consecutive days.

### 2.3. Data Collection

#### 2.3.1. Sociodemographic Factors

For each participant, the following sociodemographic factors were considered: age, sex, education, employment status, marital status, number of children, and relationship with the patient.

#### 2.3.2. Physical Activity and Inactivity Measured by Accelerometry

Study data were obtained using a GeneActiv triaxial accelerometer (Activinsights Ltd., Kimbolton, Cambs, UK), set to 40 Hz, and worn on the wrist of the non-dominant hand, 24 h a day for seven days. Data were downloaded as “bin” files using GENEActiv PC software (Activinsights Ltd., Kimbolton, Cambs, UK) and processed using the open-source R-package GGIR v.2.7-0 (R Core Team, Vienna, Austria) (https://cran.r-project.org/web/packages/GGIR/index.html) (accessed on 4, July, 2022) to determine periods of sustained inactivity and physical activity and sleep patterns. This open-source code was validated for self-calibrated functions. The sensor calibration error was minimised by auto-calibration of the data based on local gravity, as described by van Hees et al. [24].

Accelerations were determined as the Euclidean norm minus one (ENMO), i.e., √(x2 + y2 + z2) − 1G (where 1G ~9.8 m/s^2^) (negative values rounded to zero). Non-wear periods (based on the raw acceleration of the three axes, each 15 min block was classified as non-wear time if the SD of two of the three axes was <13 mG during the surrounding 60 min moving window or if the value range for two of the three axes was <50 mG) and sustained abnormally high accelerations (higher than 5.5 G for at least 15 min, related to device malfunctioning) were discounted [25]. Waking and sleeping hours were identified using an automatized algorithm [26]. Only results from participants with a wear time ≥16 h/d for at least four days (at least three weekdays and at least one weekend day) were considered valid. In each category, bouts of sustained inactivity were recorded when 80% of the threshold criteria were met.

Sustained inactivity bouts (SIB) are periods that during the night would be labelled as sleep but which during the day form a subclass of inactivity, which may represent daytime sleep or wakefulness while being motionless for a sustained period of time. During the day, SIB may refer to napping or similar sustained wrist inactivity. Sustained inactivity is detected as the absence of change in arm angle greater than five degrees for five minutes or more and is classified using the ENMO threshold of <40 mG in the non-dominant wrist [26,27,28,29]. The following SIB-related variables were considered: total duration of SIB (minutes/day), time spent in SIB in bouts of 1–10, 10–20, 20–30, and >30 min, and total number of SIBs.

For physical activity, the following variables were considered: total time of light physical activity (total LPA), defined as 40–100 mG; time spent in LPA in bouts of 1–10 min and >10 min; number of bouts of LPA; total time of moderate physical activity (total MPA), defined as 100–430 mG; total time in vigorous physical activity (total VPA), defined as ≥430 mG; total time in moderate-to-vigorous physical activity (total MVPA), defined as ≥100 mG; time spent in MVPA in bouts of 1–10 min and >10 min; number of bouts of MVPA [28,29,30,31]. Compliance with WHO criteria was scored as one if at least 150 min per week of MVPA were performed and as zero when this was not the case.

Sleep analysis was performed taking into account total time spent in bed (calculated as the difference between sleep onset and waking time), total sleep time (the accumulated nocturnal SIB during the sleep period), and sleep efficiency (the ratio between total sleep time and total time in bed) [26].

#### 2.3.3. Assessment of Caregiver’s Quality of Life

The caregiver’s quality of life was assessed using the Quality of Life Family Version (QoL-FV) instrument [32], as validated for use in Spanish [33]. This instrument was created to determine the quality of life of family caregivers of patients with cancer. It contains 37 items, scored on a scale from 0 to 10, with 10 being the best possible result. Some of these items produce an inverse score, which must be adjusted in the subsequent coding. The scale consists of four domains: physical well-being, psychological well-being, social concerns, and spiritual well-being.

#### 2.3.4. Assessment of Caregiver Strain

The Caregiver Strain Index (CSI) questionnaire [34], in the version translated and adapted to Spanish [35], was applied. This instrument measures the perceived workload of family caregivers and the effort involved in providing care. The CSI is composed of 13 items with dichotomous answer options. Each affirmative answer is scored as one. A total CSI score of seven or more represents a high level of strain, often referred to as burden of care or caregiver burden. 

Other aspects considered in assessing caregiver strain: time spent providing palliative care (months), the number of hours dedicated to daily care, and the receipt of assistance for this task. These data were obtained by personal interview.

#### 2.3.5. Ethical Considerations

This study was approved by the Malaga Provincial Ethics Committee (project code: 0638-N-18). The study participants were informed in writing and verbally about the project, and all gave signed informed consent to take part.

#### 2.3.6. Data Analysis

In the descriptive analysis performed of the study variables, the quantitative variables were expressed as mean values with the corresponding standard deviation, and the qualitative ones as the absolute frequency percentage. Relationships were determined, according to Spearman’s correlation, between the accelerometry-related variables (LPA, MPA, VPA, MVPA, inactive time, nocturnal sleeping, napping, sleep efficiency, and compliance with WHO-recommended activity criteria) and the other study variables (quality of life, CSI, daily hours spent providing care, total duration of care provision, duration of palliative care, assistance in care provision, sex, and age). All statistical analyses were performed using the SPSS 22 statistical program with *p* ≤ 0.05 considered significant and marked with an asterisk (*).

## 3. Results

### 3.1. Characteristics of the Caregivers

Initial contact was made with 83 family caregivers. However, one did not meet the inclusion criteria; in four cases, the patient in home palliative care died before the study interview was conducted; and three caregivers refused to participate. Thus, the final study sample consisted of 75 caregivers; the demographic characteristics are detailed in Table 1.

### 3.2. Descriptive Information on the Study Variables

A descriptive analysis of the variables measured by accelerometry is presented in Table 2, and strain and quality of life outcomes are presented in Table 3.

### 3.3. Bivariate Results

The following significant bivariate associations between the accelerometry variables and the other study variables were determined:-The total duration of care is associated with a greater total LPA (r = 0.231, *p* = 0.046 *) and with an LPA of 1–10 min (r = 0.259, *p* = 0.025 *).-Care provision per day is associated with greater total LPA (r = 0.230, *p* = 0.047 *).-Caregiver age is associated with lower total VPA (r = −0.236, *p* = 0.041 *) and total MVPA (r = −0.237, *p* = 0.042 *).-CSI is positively associated with an SIB of 20–30 min (r= 0.232, *p* = 0.046 *).-Total QoL-FV is associated with an SIB of 20–30 min (r = −0.322, *p* = 0.005 *).-A physical well-being subscale of QoL-FV is associated with an SIB of 20–30 min (r= −0.261, *p* = 0.024 *).-A psychological well-being subscale of QoL-FV is associated with an SIB of 20–30 min (r= −0.327, *p* = 0.04 *).-Compliance with the WHO recommendation of MVPA ≥150 min/week is associated with total QoL-FV (r = −0.269, *p* = 0.020 *).-Moreover, mean nocturnal sleep is 359.11 min (SD, 46.60) for men and 400.18 (SD, 67.11) minutes for women (*p* = 0.042 *).

All significant bivariate associations are presented in Figure 1. Other analysed outcomes did not show significant relationships.

## 4. Discussion

In our study group, most of the caregivers for patients with advanced cancer were women, usually the wife or daughter, a profile similar to that observed in previous studies of caregivers in this situation [36]. Only a third of the caregivers also worked outside the home, hence the significant time spent providing care and attention (on average, 17 h per day), as also reported in previous studies [37]. This level of commitment lasts for an average of almost 18 months, during which the caregiver is exposed to extra work, responsibility, and preoccupation, which may worsen their own mental and physical situation [38]. Only a third of these caregivers received help to carry out the tasks involved in caring for a patient with advanced cancer. Over 60% presented a high level of strain, which corroborates previous reports in this respect [39].

The caregivers’ overall quality of life was in the middle of the QoL-FEV range, which is in line with previous findings [40]. The highest scores were obtained for the physical QoL subscale and the lowest ones for spiritual well-being. From this, we conclude that care duties mainly affect the caregiver’s spiritual well-being and have much less impact on physical aspects. The above results illustrate the complexity of providing home palliative care for cancer patients, a complexity that is also present in caregivers’ daily physical activity.

A previous study, which applied multiple regression analysis, showed that depression, the self-efficacy of caregiving, the subcaregiver, and satisfaction with the home care service were associated with family caregivers’ QoL [41]. Furthermore, being female has been associated with a higher prevalence of anxiety according to multiple analyses in informal caregivers of terminally ill breast and gynaecologic cancer patients [42]. However, despite the relationship between these outcomes with the levels of physical activity and inactivity, both variables were not included in the aforementioned studies.

Moreover, multiple analyses have shown that depression and poor sleep quality did not predict physical activity levels of caregivers of patients with breast, prostate, or colorectal cancer [43]. Notwithstanding, the physical activity level and sleep quality were estimated through questionnaires [43], which do not lead to accurate measurements [44]. Thus, a strength of this study is that physical activity was assessed through an objective method consisting of accelerometry.

LPA accounted for most of the daily activity time and was positively associated with the total duration of care. This finding suggests that greater care is provided (and hence MVPA is reduced) as the disease progresses. The daily time dedicated to care was also positively associated with LPA by the caregiver. In our study group, the caregivers performed MVPA for more than 90 min a day, in line with WHO recommendations according to which at least 150 min of this activity should be performed per week. This target was met by 90% of the caregivers, and 88% performed at least 300 min a week, which confers additional health benefits [4]. 

Our findings show, therefore, that only about 10% of these caregivers present an insufficient level of physical activity, compared to international prevalence data of 27.5% [38]. The contrast is greater still with respect to data from Western countries alone (36.8%) and from women alone (42.3%) since the caregivers in our group of caregivers in Spain were mostly female [45].

However, MVPA with a minimum continuity of ten minutes was only achieved by 8% of the caregivers. From this, we conclude that the exercise performed is sufficient, overall, but the duration of MVPA is usually short, lasting less than ten minutes. Furthermore, in our group, MVPA and VPA were inversely associated with caregiver age, i.e., younger caregivers were more likely to engage in this level of activity. Thus, MVPA should be promoted in older caregivers (over 65 years old) in order to ensure active ageing.

Moreover, although the WHO recommends more than 150 min a week of MPA or 75 min a week of VPA, it is important to highlight the role of higher intensities of effort (i.e., VPA and very VPA). Along these lines, in our participants, the MVPA mostly consisted of MPA instead of VPA (94 vs. 2 min/day, see Table 2). Previous research has shown that changes in VPA are positively associated with higher cardiorespiratory fitness, but MPA showed better associations with lower body fat [46].

Some experimental designs have shown greater benefits on health outcomes with higher intensities than moderate intensities in different populations [47,48,49]. In addition, the level of MVPA has been shown to reduce the risk of cardiovascular diseases independently of the sedentary or inactive time, which is only a preventive factor in people with a low level of physical activity [50]. Therefore, the possible benefits of MVPA sustained in bouts of 10 min and the level of VPA in family caregivers are proposed according to our results.

Despite meeting WHO criteria for physical activity, the caregivers presented an average of total inactive time exceeding 11 h per day. The WHO recommends that the activity time should be raised as much as possible, even if the effort involved is only light [4]. This question requires further investigation to determine the causes; for example, it may be due to physical fatigue or because the caregiver is unable to leave the patient alone or with anyone else.

Among other bivariate associations detected, our analysis shows that caregivers who present more periods of inactivity lasting 20–30 min tend to experience higher levels of strain. We also found that caregivers with fewer moments of inactivity lasting 20–30 min had a better quality of life, overall, and reported better physical and psychological well-being. Therefore, caregiver strain and poorer quality of life seem to be more related to the inactivity time than to the physical activity level.

According to the study results, the majority of caregivers comply with the WHO recommendations on physical activity. However, although providing this type of home care for cancer patients requires a certain degree of MVPA, the question arises as to whether this activity produces the same physical and mental benefits as other types of activity, such as walking, dancing, or practicing sports. The situation of caregivers is often complicated. On the one hand, there is known to be an association between a better quality of life and less compliance with WHO criteria for physical activity. However, on the other hand, with a better quality of life there are fewer periods of inactivity lasting 20–30 min.

This study assesses important aspects for the prevention of health problems in family caregivers in relation to physical activity and inactivity time. To enable caregivers to reduce their inactivity, strain must be alleviated and the quality of life improved. The WHO recommendations for physical activity could be applied with caution in this population; therefore, public health actions must adapt the recommendations making them more specific since family caregivers with a better quality of life are less likely to meet WHO guidelines. Those caregivers who comply with the WHO guidelines of PA are probably achieving it by a higher care time, showing higher caregiver burden, and lower self-reported quality of life. Strategies to reduce inactivity time are encouraged, especially with ageing, in which MVPA levels are reduced, compromising the healthy ageing of the family caregiver. 

The study data reflect the need to develop strategies enabling family nurses to enhance the care and health status of caregivers. Such an advance would also benefit the patients. However, this study has some limitations since, as we described, it is cross-sectional in nature, and therefore causality cannot be inferred. In addition, the outcomes observed in the study variables may be influenced by aspects other than those considered; in fact, the strength of the mentioned associations was limited with r coefficients of 0.2–0.3. Experimental designs are encouraged with a higher simple size of caregivers of palliative cancer patients, despite the difficulties due to the little free time of this population.

## 5. Conclusions

Most of the caregivers in our study sample presented a high level of physical activity, in line with WHO recommendations, possibly due to the requirements inherent in caring for a terminal patient. However, the total duration of inactivity was also high, at an average of over 11 h per day, which increases the risk of health problems. Inactivity periods of 20–30 min are slightly associated with greater caregiver burden and a poorer quality of life. Our findings also show that family caregivers with a better quality of life are less likely to meet the WHO criteria for physical activity, with slight associations. Older caregivers achieved lower levels of MVPA than the younger ones, highlighting the needed for active ageing also in this population in which a large part of the time is dedicated to care.

## Figures and Tables

**Figure 1 ijerph-20-00179-f001:**
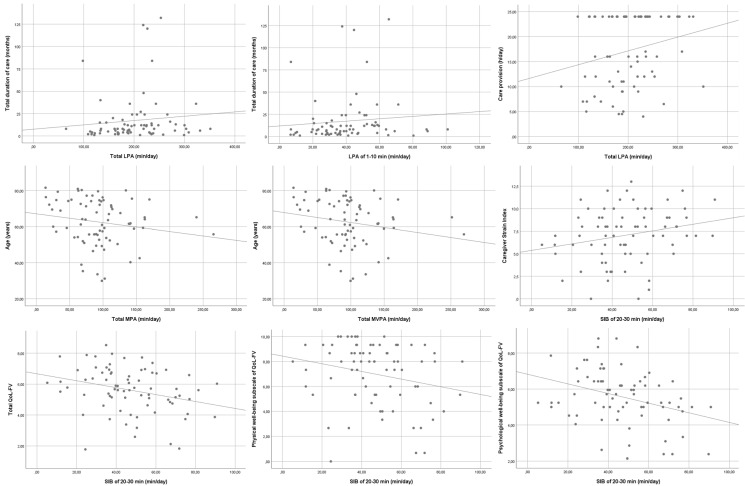
Significant bivariate associations.

**Table 1 ijerph-20-00179-t001:** Demographic characteristics of the family caregivers (n = 75).

	Mean	SD
**Age (years)**	62.71	12.80
	**N**	**%**
**Gender**		
Female	66	88.0
Male	9	12.0
**Relationship with patient**		
Spouse	41	54.7
Daughter	25	33.3
Sister	4	5.3
Mother	2	2.7
Daughter-in-law	1	1.3
Sister-in-law	1	1.3
Mother-in-law	1	1.3
**Caregiver’s marital status**		
Married	52	69.3
Single	12	16.0
Divorced	9	12.0
Widowed	2	2.7
**Caregiver’s children (n)**		
None	15	20.0
One	10	13.3
Two	29	38.7
Three	17	22.7
Four or more	4	5.3
**Caregiver’s education**		
Primary studies	39	52.0
Secondary studies	20	26.7
No formal studies	8	10.7
University studies	8	10.7
**Employment status**		
No paid employment	36	48.00
In paid employment	26	34.67
Retired	13	17.33

SD: standard deviation.

**Table 2 ijerph-20-00179-t002:** Accelerometry outcomes in the family caregivers (n = 75).

	Mean	SD
SIB total (min/day)	665.78	94.92
SIB 1–10 min (min/day)	69.40	25.93
SIB 10–20 min (min/day)	57.11	21.88
SIB 20–30 min (min/day)	46.84	18.91
SIB >30 min (min/day)	500.43	155.20
Total LPA (min/day)	201.64	59.06
LPA 1–10 min (min/day)	40.34	19.73
LPA >10 min (min/day)	4.24	10.59
Total MPA (min/day)	93.92	45.63
Total VPA (min/day)	2.47	5.46
Total MVPA (min/day)	96.40	46.93
MVPA 1–10 min (min/day)	26.14	19.00
MVPA >10 min (min/day)	9.48	15.95
Napping (min/day)	70.15	64.11
Nocturnal sleep (min/day)	395.25	66.12
Sleep efficiency	0.84	0.09
	**N**	**%**
Compliance with WHO recommendation (MVPA ≥150 min/week) in episodes of at least 1 min/day		
Yes	68	90.7
No	7	9.3
MVPA (≥150 min/week) in episodes of at least 10 min/day		
Yes	6	8.0
No	69	92.0
MVPA (≥300 min/week) in episodes of at least 1 min/day		
Yes	66	88.0
No	9	12.0

SD: standard deviation, SIB: sustained inactivity bouts, LPA: light physical activity, MPA: moderate physical activity, VPA: vigorous physical activity, MVPA: moderate-to-vigorous physical activity, and WHO: World Health Organization.

**Table 3 ijerph-20-00179-t003:** Strain and quality of life of the family caregivers (n = 75).

	Mean	SD
**Quality of Life Family Version**		
Physical domain	6.99	2.52
Social domain	5.92	1.97
Psychological domain	5.57	1.5
Spiritual domain	3.98	1.85
Total quality of life	5.62	1.46
**Care provision (hours/day)**	17.19	7.18
**Duration of care (months)**	17.33	27.04
**Duration of palliative care (months)**	3.93	4.82
	**N**	**%**
**Assistance received for care provision**		
No	48	64.0
Yes	27	36.0
**Caregiver Strain Index**		
≥7 points: 46 61.3	46	61.3
<7 points: 29 38.7	29	38.7

SD: standard deviation.

## Data Availability

Data available under reasonable request to the corresponding author.

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
