# Peer review of "Accelerometer-Measured Physical Activity, Inactivity, and Related Factors in Family Caregivers of Patients with Terminal Cancer"

_ijerph, 2022, doi:10.3390/ijerph20010179_

Round 1
Reviewer 1 Report
To whom it may concern,
The authors wrote the manuscript really well with all the necessary information. There are a couple of things that need to be changed, as I mentioned in the comment section of the article. Here I attach the article with my comments.
Thank you,
Kushal Gandhi.

Author Response
Response to Reviewer 1 Comments
Point 1: To whom it may concern,
The authors wrote the manuscript really well with all the necessary information. There are a couple of things that need to be changed, as I mentioned in the comment section of the article. Here I attach the article with my comments.
Thank you,
Kushal Gandhi.
Response 1: Dear Kushal Gandhi, thank you very much for your comments. All the suggestions of the attached document have been made.
Point 2: Please add asterisk (*) if you have significant value.
Response 2: We have added asterisks when p<0.05, thus, our results are clearer presented. Thank you.
Point 3: Add comma after responsibility.
Response 3: Corrected.
Point 4: Instead of deduce, conclude would be better fit in the sentence.
Response 4: Corrected.
All the best,
Cristina Casals.

Reviewer 2 Report
The authors evaluated the relationship between caregiver burden or quality of life and daily physical activity and inactivity measured by an accelerometer. There are some concerns that should be addressed.
1. Enough sample size is necessary to reach a significant result. Sample estimation should be performed first.
2. Scatter plot should be presented for every pair of related variables.
3. It is better to list the bivariate result in a table.
4. Most of the correlation coefficients are <0.3, the relationship is not strong enough to derive the conclusion.
5. multiple variables analysis may be a more appropriate method.
Author Response
Response to Reviewer 2 Comments
The authors evaluated the relationship between caregiver burden or quality of life and daily physical activity and inactivity measured by an accelerometer. There are some concerns that should be addressed.
Point 1: Enough sample size is necessary to reach a significant result. Sample estimation should be performed first.
Response 1: We have detailed this aspect according to the Reviewer’s suggestion. Moreover, we have included in limitations the suitability of a higher sample size, however, it is important to highlight the difficulty of including a specific sample (informal caregivers of cancer patients in palliative care).
Indeed, the sample size calculation was not included in our first draft due to the lack of studies on physical activity in informal caregivers of cancer patients in palliative care. We were not sure about its suitability.
When we prepared this funded project, we calculated our sample considering caregivers of Alzheimer's patients as “a similar population”, obtaining a sample size can of 75 caregivers. To calculate the necessary sample size, the data from Connell & Janevic, 2009 were used. These authors, studying the physical activity performed by caregivers, spouses of patients with Alzheimer, found that their sample at the beginning of the study performs 5.2 hours/week of physical activity with a standard deviation of 2.2 hours/week. With these data, a sample of 75 participants was estimated as necessary for an alpha error of 0.05 and an estimation error of 0.5. Sample size analyses have been calculated using the Epidat 4.2 program.
Connell, C.M., & Janevic, M.R. (2009). Effects of a Telephone-Based Exercise Intervention for Dementia Caregiving Wives: A Randomized Controlled Trial. Journal of applied gerontology: the official journal of the Southern Gerontological Society, 28(2), 171–194. https://doi.org/10.1177/0733464808326951
According to one of our main associations, the relationship between quality of life and SIB 20-30 and using the G*Power software (v. 3.1.9.6, University of Kiel, Germany), we obtained a compute achieved power of 0.82. Similarly, the association between quality of life and total MVPA showed a power of 0.86. Thus, sample size seems to be enough to these analyses, although a statement has been included in limitations regarding sample size.
Point 2: Scatter plot should be presented for every pair of related variables.
Response 2: We agree with the reviewer, therefore, we have included all scatter plots in Figure 1 and it has been cited in the manuscript. Thanks to the inclusion of these figures, as suggested by Reviewer 2, the correlation analyses are presented in a more appropriate way since outliers or atypical distributions could be detected.
Point 3: It is better to list the bivariate result in a table.
Response 3: Although this option proposed by Reviewer 2 is valid, we have included 3 tables and we prefer not to overload the reader with excessive tables; in addition, we also have to attend to the requests of Reviewer 1. Thus, we have presented the bivariate associations in Text, but also in Figures, in the hope that it has been sufficiently improved and will please both reviewers and, in case of publication, the readers.
Point 4: Most of the correlation coefficients are <0.3, the relationship is not strong enough to derive the conclusion.
Response 4: Thank you very much for you detailed review. We have emphasized this fact in the text appropriately and, consequently, we have rewritten the limitations and conclusions. We agree with the Reviewer 2. All changes have been highlighted in yellow in the revised manuscript. Our correlations coefficients can be low due the importance of other factors (functional dependency of the patient, help received, employment status, etc.), however, we still consider these associations of interest since PA can be a modifiable lifestyle factor. In this line, although the WHO provides PA recommendations, they are for the general population and the caregivers, and the informal caregivers of palliative cancer patients, must be specifically analysed.
Point 5: Multiple variables analysis may be a more appropriate method.
Response 5: Both bivariate and multivariate analyses are statistical methods to investigate relationships between data samples. In our first draft, we applied bivariate analysis since it looks at two paired data sets, studying whether a relationship exists between them, giving a “clear” and “honest” result without adjustments, etc. As multivariate analysis examines several variables to see if one or more of them are predictive of a certain outcome, the predictive variables are independent variables and the outcomes are the dependent variables which should not be related.
According to the Reviewer suggestion, we have completed our analyses providing multiple variables analyses. Specifically, multiple linear regressions with outcomes from accelerometry (physical activity outcomes, inactivity outcomes, and sleep variables), and using the stepwise method, were applied to predict CSI, total QoL-FV, and each of the five dimensions of QoL-FV, showing the following results:
- The model with CSI only included SIB of 20-30 minutes (standardized β = 0.26, p= 0.03). Also, it can be proposed the prediction of CSI with SIB of 20-30 minutes (standardized β = 0.31, p= 0.01) and LPA 10 min (standardized β = -0.24, p= 0.04).
- The model with total QoL-FV only included SIB of 20-30 minutes (standardized β = -0.29, p= 0.01).
- Psychological domain of QoL only included SIB of 20-30 minutes (standardized β = -0.34, p= 0.03). Physical scale of QoL only included SIB of 20-30 minutes (standardized β = -0.23, p= 0.04). No other significant relationships were found with the other 3 domains of QoL.
Therefore, the results provided by multiple linear regressions are similar to our first analysis (with bivariate correlations) and, in the proposed model, we are introducing several non-independent and related variables what is not appropriate. We would like to clarify that theses variables are related since the higher PA a person performs, the lower time to stay sedentary in a day. All variables are providing information about activities you perform in a day, when one of them increases, other must decrease.
Thus, we are not sure about the suitability of this analysis and we are only including this information in the response to reviewers, but not in the revised manuscript. In case that the Reviewers consider to include it in the final manuscript, just let us know and we will kindly include it.
__________________________________________________
Thank you very much to both Reviewers for helping us to improve our study.

Round 2
Reviewer 2 Report
The authors could add some articles including multiple analysis in the discussion part.
Author Response
Comment 1: The authors could add some articles including multiple analysis in the discussion part.
Response 1: Five new references have been added in the discussion which applied multiple analyses. Thank you very much for your help during this process.
All the best,
Cristina